# An Inverted Honeycomb Plasmonic Lattice as an Efficient Refractive Index Sensor

**DOI:** 10.3390/nano11051217

**Published:** 2021-05-04

**Authors:** Javier Rodríguez-Álvarez, Lorenzo Gnoatto, Marc Martínez-Castells, Albert Guerrero, Xavier Borrisé, Arantxa Fraile Rodríguez, Xavier Batlle, Amílcar Labarta

**Affiliations:** 1Departament de Física de la Matèria Condensada, Universitat de Barcelona, 08028 Barcelona, Spain; lgnoatgn26@alumnes.ub.edu (L.G.); martinezcastellsmarc@gmail.com (M.M.-C.); arantxa.fraile@ub.edu (A.F.R.); xavierbatlle@ub.edu (X.B.); amilcar.labarta@ub.edu (A.L.); 2Institut de Nanociència i Nanotecnologia (IN2UB), 08028 Barcelona, Spain; 3Institut de Microelectrónica de Barcelona (IMB-CNM, CSIC), 08193 Bellaterra, Spain; albert.guerrero@imb-cnm.csic.es; 4Catalan Institute of Nanoscience and Nanotechnology (ICN2), CSIC and The Barcelona Institute of Science and Technology, Campus UAB, Bellaterra, 08193 Barcelona, Spain; xavier.borrise@imb-cnm.csic.es

**Keywords:** Au plasmonic nanostructures, inverted honeycomb lattice, surface lattice resonances, refractive index sensor

## Abstract

We present an efficient refractive index sensor consisting of a heterostructure that contains an Au inverted honeycomb lattice as a main sensing element. Our design aims at maximizing the out-of-plane near-field distributions of the collective modes of the lattice mapping the sensor surroundings. These modes are further enhanced by a patterned SiO_2_ layer with the same inverted honeycomb lattice, an SiO_2_ spacer, and an Au mirror underneath the Au sensing layer that contribute to achieving a high performance. The optical response of the heterostructure was studied by numerical simulation. The results corresponding to one of the collective modes showed high sensitivity values ranging from 99 to 395 nm/RIU for relatively thin layers of test materials within 50 and 200 nm. In addition, the figure of merit of the sensor detecting slight changes of the refractive index of a water medium at a fixed wavelength was as high as 199 RIU^−1^. As an experimental proof of concept, the heterostructure was manufactured by a simple method based on electron beam lithography and the measured optical response reproduces the simulations. This work paves the way for improving both the sensitivity of plasmonic sensors and the signal of some enhanced surface spectroscopies.

## 1. Introduction

The design and manipulation of plasmonic nanostructures have rapidly become one of the most active topics in nanophotonics over the last years. Specifically, several plasmonics-based approaches have been applied to improve the detection limit and sensitivity of sensing devices. Good examples of these new capabilities are the potential to prove volumes beyond the diffraction limit [1,2,3] or to achieve single-molecule sensitivity [4]. Moreover, plasmonic nanostructures have contributed to largely enhancing the signal of surface spectroscopies, such as surface-enhanced Raman scattering or plasmon-enhanced fluorescence [5,6,7,8,9].

One of the most promising niches for plasmonic nanostructures is their integration in refractive index sensors [10]. This kind of sensor benefits from the fact that the wavelengths at which the plasmonic excitations arise depend strongly on the features of the medium mapping the corresponding near-field distributions. However, the inherent nature of localized surface resonances (LSRs) hinders their efficiency as sensing devices as the evanescent electric fields excited in an LSR decrease rapidly with the distance from the metal–dielectric interface. In this way, the regions in which the electric field is enhanced, the so-called “hotspots”, extend only a few tens of nanometers away from the metal–dielectric interface and, in many cases, are confined in the space between neighboring plasmonic elements. In contrast, inverted structures, in which dielectric holes are carved through a continuous metallic layer, have proven to be a valid approach to tackle this issue [11] and are known to improve the efficiency of light–matter interaction in some situations [12,13,14,15,16]. Unlike other realizations of inverted plasmonic structures [17,18,19], such as nanoholes or nanoarrays with a more complex pattern, the heterostructure proposed in this work benefits from the inclusion of a mirror layer. By adding this element, the device can collect all the optical responses solely in the reflection spectrum, given the null transmission.

In addition, our system is based on the detection of surface lattice resonances (SLRs) in a honeycomb lattice, which implies sharper peaks associated with higher-energy modes involving the collective excitation of all the elements in the plasmonic array [20,21,22,23]. These modes usually present hotspots encompassing larger areas than those associated with local excitations. Besides, the three-fold symmetry of the honeycomb lattice may hamper the excitation of LSR in favor of SLR, making collective modes much more intense [20,21], as will be further discussed in Section 3.2.

Overall, the heterostructure presented in this work constitutes a proof of concept for a simple realization of a highly sensitive refractive index sensor that can be manufactured by a relatively easy lithographic process.

## 2. Materials and Methods

As shown in Figure 1, the main part of the studied heterostructure is composed of two layers of Au and SiO_2_ of 30 and 80 nm in thickness, respectively, where an inverted honeycomb lattice—with a pitch of 866 nm and bar-shaped trenches of 400 nm in length and 30 nm in width—is carved through the whole Au/SiO_2_ bilayer. On the bottom of the trenches, there is an Au layer of 30 nm in thickness that shapes the direct honeycomb lattice complementary to that at the top. Thus, the two plasmonic honeycomb lattices are placed at an edge-to-edge distance of 50 nm. Between them, there is a sort of two-dimensional photonic crystal that is formed by the contrast in the refractive indices of the SiO_2_ and the material filling the trenches (air, water, and so on). Consequently, the interactions among the three lattices feed back into each other because they share the same pitch and symmetry. Specifically, one of the collective modes of the two plasmonic structures gives rise to a spatial distribution of the near-field enhancement around the heterostructure that is well suited for sensing applications, as discussed later. This heterostructure rests on top of a second bilayer, composed of an SiO_2_ spacer and an Au mirror of 120 nm and 100 nm in thickness, respectively, which enhances the intensity of the collective modes and ensures that the transmission through the entire system is negligible.

The simulations presented in this work were performed using the finite-difference time-domain (FDTD) method, implemented in the solver provided by Lumerical^R^ [24].The method defines a simulation spatial domain, which is discretized. Then, the incident electromagnetic field is set by defining the initial state of the system. The software solves Maxwell’s equations to determine the evolution of the electric and magnetic fields using a discrete time-step. This approximation, as well as the inherent mesh step necessary to discretize the space in the simulation domain, introduce a certain error into the calculations. However, the method allows for direct observation of the physical phenomena taking place without imposing any further assumption on the behavior of the system.

The configuration of the simulations was as follows: a short pulse of linearly polarized light was sent perpendicularly to the surface of the heterostructure from a source placed above the metal (see the inset of Figure 1b). Placing different monitors in several spots, we obtained the transmission and reflection cross sections, as well as the electric field distribution as a function of the wavelength of the incident radiation. Owing to the periodicity of the system, periodic boundary conditions of a rectangular unit cell of the honeycomb lattice are used to perform the simulations (see Figure 1b). The dielectric functions for the materials used in these simulations were obtained by fitting analytical functions to the data from [25] for SiO_2_ and Au, respectively.

In the manufacturing of the samples, a bilayer made up of Au (mirror) and SiO_2_ (spacer) layers of 100 nm in thickness each was deposited on top of a standard Si wafer by electron beam evaporation (ATC Orion, AJA International, Inc) and plasma enhanced chemical vapor deposition (PECVD), respectively. A pattern of 500 µm × 500 µm in size following the inverted honeycomb design shown in Figure 1 was exposed by electron beam lithography (EBL) on an additional third layer of poly methyl methacrylate (950 PMMA, MicroChem) with a thickness of 90 nm. After removing the exposed areas, the sample was metallized with 30 nm of Au deposited by electron beam evaporation. The outcome of the evaporation process was a 30 nm Au layer on top of the heterostructure shaping an inverted honeycomb lattice and the complementary direct version of the same lattice on the bottom of the trenches, as shown in Figure 1a, at an edge-to-edge distance of 60 nm. Between them, a two-dimensional photonic crystal is formed by the contrast in the refractive indices of the 950 PMMA and the material filling the trenches. No further lift-off was necessary owing to the design of the samples. As SiO_2_ and 950 PMMA have very similar refractive indices, the optical spectra of the simulated and manufactured heterostructures may be compared right away, even though some geometrical parameters are slightly different than in the simulations, namely, the edge-to-edge distances between the inverted Au layer and both the mirror (190 nm) and the direct Au structure (60 nm).

The optical characterization was carried out by a Vertex 70 Fourier transform infrared (FTIR) spectrophotometer attached to an optical microscope (Bruker Hyperion). Experiments were performed in the reflection configuration with a 4× objective and under the illumination of unpolarized light. The measured signal of an Ag mirror was used as a background in all the experiments.

## 3. Results

### 3.1. Optical Response of the Heterostructure

Several peaks are exhibited by the simulated reflection spectrum of this heterostructure (see Figure 2). Nevertheless, they can be classified into only two groups depending on either the local or collective nature of the resonances associated with them. The first group includes the peaks located around 729 and 1789 nm, which arise from the excitation of LSR related to local modes of the bar-shaped trenches and the Au bars in the inverted and direct plasmonic structures, respectively. Accordingly, the near-field distributions for these modes are mainly confined inside either the trenches through the Au layer or around the Au bars at the bottom of the SiO_2_ trenches. For instance, the corresponding near-field distributions for these two peaks show mostly dipolar excitations of the trenches perpendicular to the polarization axis of the incoming radiation with some multipolar or dipolar polarizations of the tilted trenches for the peaks at 729 and 1789 nm, respectively (see Appendix A).

It is also worth noting that the positions of these two peaks can be tuned just by adding the Au mirror and varying the edge-to-edge distance between the inverted Au lattice and the Au mirror as there is a certain coupling with the images of these modes at the mirror. In such a way, 200 nm was chosen for the edge-to-edge distance so that the peak at 729 nm was blue-shifted to be on the left-hand side of the sharp peak at 765 nm, which is the one we are intending to use for sensing applications. In a similar way, the edge-to-edge distance between the inverted Au lattice and the direct one was optimized to be 50 nm to reduce the overlap between the peaks at 729 and 765 nm.

In the second group of excitations shown in Figure 2, there are those with a collective character that are associated with SLRs. The corresponding peaks are the aforementioned one at 765 nm and the ones located about 754 and 1107 nm. These three peaks arise from the constructive interference between the incident radiation and the collective modes of the direct and inverted honeycomb plasmonic lattices. Considering that SLRs are excited at the interface between the Au and the surrounding dielectric media, two SLRs should be expected for each honeycomb lattice as both the direct and the inverted Au arrays are in contact with two different dielectric media, namely air (above) and SiO_2_ (below). For a hexagonal lattice, which is the Bravais lattice of the honeycomb array, the condition for the excitation with perpendicular incidence of the most intense SLR is
(1)$$\lambda_{i}=n_{i}\;p\cos30,$$
where $\lambda_{i}$ is the resonant wavelength of the incident radiation, p is the pitch of the hexagonal lattice (p= 866 nm; see Figure 1b), and $n_{i}$ is the effective refractive index of the dielectric medium forming the interface with the Au layer. Taking average values of the refractive indices of air and SiO_2_ as 1 and 1.45, respectively, the expected wavelengths that fulfill Equation (1) are 750 nm and 1087 nm, respectively, in very close agreement with the positions of the peaks found by numerical simulation.

Thus, the small peak around 754 nm is mainly related to the SLR at the interface with the air of the direct honeycomb lattice on the bottom of the SiO_2_ trenches. This is supported by both the near-field distribution shown in Appendix A and the fact that this peak does not appear when the direct Au honeycomb lattice is not present, and the trenches are carved through the whole thickness of the SiO_2_ layer down to the Au mirror (see Appendix A). Interestingly, in the latter case, the peak around 729 nm remains almost unchanged, strengthening the argument that it is only associated with local modes of the trenches in the inverted Au lattice. The second SLR mode corresponding to the Au–SiO_2_ interface of the direct honeycomb lattice may be likely overlapped with the intense peak around 1107 nm.

On the contrary, the two peaks around 765 and 1107 nm are associated with the two SLRs at the air–Au and Au–SiO_2_ interfaces of the inverted honeycomb lattice, respectively. Both peaks are very intense and relatively sharp, which makes them suitable for applications. It is also worth noting that the overlap between the two peaks at 729 and 765 nm is somehow reduced by the coupling with the two-dimensional photonic crystal underneath the inverted plasmonic lattice, which is beneficial for sensing purposes. As a comparison, Appendix A shows the spectra for the cases of a continuous SiO_2_ spacer of 200 nm in thickness and the same spacer, but with trenches carved through its entire thickness.

In addition, simulations of the near-field distributions on the heterostructure at 765 and 1107 nm reveal the formation of patterns of extended hotspots compatible with the translational symmetry of the hexagonal lattice. For the peak at 1107 nm, there are also some extra excitations inside the trenches of the inverted Au layer (Figure 3b) arising from the coupling between the SLR modes of the direct and the inverted lattices through the polarization of the regions of the trenches in between the two plasmonic lattices (see Figure 3d). Although the two peaks show very similar near-field patterns outside the trenches, it is interesting to note that, for the peak at 765 nm, the hotspots following the lattice symmetry are found on top of the plasmonic Au layer, whereas in the case of the peak at 1107 nm, the electric field enhancement takes place at the bottom interface with the SiO_2_ layer (see Figure 3c,d). In both cases, the pattern of hotspots yields an out-of-plane electric field that extends hundreds of nanometers away from the corresponding interfaces. The corresponding distributions of the electric field enhancement spread deep inside the air and the SiO_2_, respectively (see Figure 3c,d). It is precisely these out-of-plane near-field distributions associated with the SLR of the inverted honeycomb lattice that provide the heterostructure with strong sensing capabilities, as will be discussed in the following subsection.

### 3.2. Performance of the Refractive Index Sensor

The design of the plasmonic heterostructure following a lattice with three-fold symmetry hampers the excitation of the LSRs that are not fully compatible with the symmetry of a hexagonal lattice, because local modes imply polarizations with an even number of charge poles [20,21]. For instance, the three bar-shaped elements of the honeycomb lattice converging on each corner of the hexagons are not compatible with a simple dipolar mode within the gap between them. This favors the excitation of the SLRs yielding more intense absorption peaks, which in turn may improve the optical response of the system for sensing purposes. While preserving the honeycomb design, the system presented in this work does not follow direct realizations of a honeycomb lattice based on arrays of plasmonic nanoelements [26,27,28]. Instead, it takes advantage of the inverted version of the direct lattice. The main plasmonic structure presented here is, as opposed to a honeycomb array of bars, a continuous layer of Au where bar-shaped trenches are carved through its full thickness. According to Babinet’s principle, the direct and the inverted structures should be equivalent in terms of their optical properties [14,29], showing the same excitations at the same values of the wavelength, but the nature of the excitations is different in the two structures. The electric dipoles formed in the direct structure are magnetic dipoles in the corresponding inverted realization and vice versa, because the role of the charge accumulation is played by the currents induced in the system in the complementary structure. Previous works show that, despite the similarities in the spectral behavior, inverted structures can perform better in terms of their interaction with the surrounding medium or an externally exciting source [7,8,9]. One particularly appealing characteristic of the inverted structures is that the near-field distributions tend to spread further away compared with the direct structure. The near-field distributions for the peaks at 765 and 1107 in Figure 3 are good examples of this spreading. In particular, the SLR at about 765 nm exhibits hotspots that extend out-of-plane largely away from the top Au–air interface. Then, any changes in the properties of the surrounding medium above the heterostructure will especially affect this excitation because the electric field is distributed in a relatively large volume that extends up to about 400 nm from the Au–air interface (see Figure 3c,d). This allows both to detect slight changes in the optical properties of a relatively distant region of the medium above and to boost the sensitivity of surface enhanced spectroscopies, such as Raman scattering.

In this work, we aim at presenting a sensing application based on the detection of small changes in the refractive index of the medium above the heterostructure by measuring the shift in the wavelength of the SLR at the interface with the top Au layer. More precisely, a 50 nm layer of a test material with a given refractive index is placed on top of the heterostructure. This layer is intended to model the adsorption on top of the heterostructure of some biological species, such as macromolecules or viruses (with refractive index values in the range of our test layer) or any other materials to be detected not fitting inside the trenches because of their relatively small width (30 nm). We decided to use a thin test layer to evidence the high sensitivity of the sensor. In Figure 4, the reflection spectra around the two SLR peaks of the inverted plasmonic structure are depicted for values of the refractive index of the test layer ranging from 1 to 1.45. Although the two peaks show remarkable shifts, the one for the SLR at the top test layer–Au interface is much larger, as expected from the corresponding near-field distributions shown in Figure 3. The fact that the near-field enhancement for this excitation is mostly distributed outside of the heterostructure makes its wavelength highly susceptible to changes in the surrounding medium.

It is worth noting that the shift in the wavelength of this resonance follows an almost perfectly linear dependence on the refractive index of the test layer (see Figure 5a). This can be easily understood considering that the resonant wavelength satisfying a SLR is linearly dependent on the effective refractive index of the dielectric medium that forms the interface with the Au layer, as shown in Equation (1). Hence, the sensitivity of the sensor computed from the slope of the linear dependence shown in Figure 5a is 99 nm/RIU (RIU stands for refractive index unit), which is a very remarkable result considering the small thickness of the test layer.

To determine the dependence of the sensitivity on the amount of test material, the same study was repeated for sensing layers of 100 and 200 nm in thickness, obtaining sensitivities of 216 and 395 nm/RIU, respectively (see Figure 5a). These results show an almost linear dependence of the sensor sensitivity on the thickness of the test material, at least in this range of relatively small thicknesses. This enables to quantify the amount of material deposited on the sensor provided its refractive index is known. In Figure 5b, the wavelength of the SLR is depicted as a function of the thickness of the test layer. For a thickness larger than 400 nm (560 nm/RIU), the sensitivity significantly departs from the linear dependence tending to a maximum value slightly greater than that of the 600 nm layer (613 nm/RIU). Therefore, the sensitivity for test layers thicker than about 1000 nm can be considered thickness independent.

The performance of the heterostructure as a detector of an analyte present in a water medium was also tested. For this purpose, water was modelled by a 600 nm thick layer with a refractive index of 1.33 that also filled the bar-shaped trenches of the inverted honeycomb lattice. The presence of an analyte was simulated by changing the refractive index of the medium to 1.34. To ensure high sensitivity to such small changes of the refractive index, the variation of the intensity of the reflection peak at a given wavelength was measured, instead of the peak shift as in the previous cases, as the simplicity in detecting a signal at a fixed wavelength is of key importance for biosensing. The figure of merit (FOM) [30] for this kind of measurements is defined as follows:(2)$$FOM=\frac{dI/dn}{I_{m}},$$
where dI is the variation in the intensity of the reflection peak corresponding to the SLR at the air–Au interface as a function of the wavelength, n is the refractive index of the medium, and $I_{m}$ is the average of the intensities of the two spectra used in the calculation as a function of the wavelength. Thus, the shift in the position of the SLR peak as n varies is translated into a change of its intensity for a fixed wavelength. The computed values of the FOM as a function of the wavelength are shown in Figure 6 together with the two spectra used in the calculations. The maximum value of the FOM is 199 RIU^−1^ at a wavelength of 974 nm. This value can be compared with the sensitivity computed from Figure 4 (613 nm RIU^−1^ for a test layer of 600 nm in thickness) divided by the full width at half maximum of the peak, which is about 9 nm [31]. Finally, the minimum detectable change in the refractive index can be estimated. For a fixed wavelength of 981 nm, for which the FOM is maximized, there is a change of 0.48 in the reflection signal induced by a change of 0.01 RIU in the refractive index (see Figure 6). Considering a minimum detectable variation of 0.1 in the signal intensity, the minimum change in the refractive index that could be detectable by the system would be 0.002 RIU. These results are a solid proof of concept for further applications of these heterostructures for biosensing purposes.

### 3.3. Experimental Realization of the Heterostructure

In addition to the results of the simulations shown throughout this article, a preliminary experimental realization and optical characterization of the heterostructure were also carried out. We aimed at replicating as much as possible the design studied in the simulations while keeping in mind the ease and potential scalability in the manufacturing process. Hence, the parameters of the modelled system were followed, apart from the nanostructured layer of SiO_2_ that was substituted by a PMMA layer with a thickness of 90 nm, giving an edge-to-edge distance between the inverted honeycomb lattice and the Au mirror of 190 nm instead of 200 nm, as was the case of the simulations. In this way, the PMMA layer was etched following a much more direct and easy lithographic method than would have been necessary for the SiO_2_ layer, while preserving an almost equal value of the refractive index. The final metallization of the heterostructure yielded the Au inverted honeycomb lattice on top of the PMMA layer, but also an additional direct lattice on the bottom of the bar-shaped trenches carved through the PMMA layer. This was the reason the Au direct lattice was also included in the simulated model, because it is hard to avoid its formation following a simple manufacturing process. Nevertheless, the presence of this extra Au direct lattice does not significantly affect the collective excitations of the inverted Au layer, as proven by the simulation of the system without the direct lattice shown in Appendix A, while it hinders the excitation of the LSR at 729 nm by partially shielding the interaction of this mode and its corresponding image at the mirror (compare the intensity of this peak in the spectra in Figure 2 and Appendix A). Figure 7a shows an scanning electron microscopy (SEM) top-view image of the final heterostructure from which the overall good quality of the manufactured sample can be assessed.

The optical characterization of the manufactured sample by FTIR spectroscopy showed the excitation of the two collective modes associated with the two interfaces of the inverted honeycomb lattice, as shown in Figure 7b. The peak corresponding to the SLR excited in the air–Au interface was broader than that of the SLR in the Au–PMMA interface. Experimental imperfections on the manufactured sample together with the existence of other excitations in the vicinity of the peak around 765 nm may cause their overlap, giving rise to a single broad peak. Nevertheless, the measured spectrum and the simulation results were in good agreement, thus proving the experimental feasibility of this heterostructure.

## 4. Discussion

The results shown in this work evidence the high performance of the presented heterostructure based on an Au inverted honeycomb lattice for sensing purposes. Owing to the chosen lattice pitch and because local excitations are hindered by the three-fold symmetry of the honeycomb array, this heterostructure exhibits very intense and narrow reflection peaks corresponding to its collective excitations at wavelengths in the near infrared that are well suited for sensing applications. The intensity of these peaks is further enhanced by the underneath layers within the heterostructure, namely, the patterned SiO_2_ layer with the inverted honeycomb lattice, the SiO_2_ spacer, and the Au mirror that ensures total reflection. Among the collective excitations shown by this heterostructure, the most suitable for sensing the refractive index of the environment is that associated with the peak around 765 nm. This peak is particularly susceptible to changes in the nearby environment because it arises at the top interface of the sensing Au layer, thus yielding an out-of-plane near-field distribution that spreads up and far away from the interface. The sensitivity associated with the shift of this peak was assessed by changing the refractive index of a thin layer of a test material resting on top of the heterostructure. Under these conditions, simulations showed a sensitivity of 99 nm/RIU for a layer of 50 nm in thickness with a full width at half maximum of the peak of only 9 nm. These results become especially relevant when compared with those of previous studies in which the refractive index change was tested in layers of infinite thickness [32,33]. Moreover, a further large shift in the resonant wavelength was also found for an increasing thickness of the test layer. This yielded a rising sensitivity for thicker test layers, opening another possible application of the studied heterostructure as a sensor of the amount of test material, provided its refractive index is known. The sensitivity of the sensor tended to a maximum value greater than about 613 nm/RIU for a thickness larger than about 600 nm, which is a very remarkable performance for this kind of sensor. Finally, the sensitivity of the sensor detecting slight changes of the refractive index at a fixed wavelength of a thick layer of a water medium was also checked. The computed FOM at 974 nm was as high as 199 RIU^−1^. This value constitutes a solid improvement over other plasmonic sensors operating in a similar fashion that exhibit values of the FOM about one order of magnitude smaller [30,33,34,35].

As an experimental proof of concept, the heterostructure was manufactured by a simple method based on EBL. The FTIR spectrophotometry measurements showed intense and relatively narrow excitations around the wavelengths predicted by the simulations. These results show that an easy experimental realization of this heterostructure is possible, in a similar way to in [22], but avoiding the final lift-off process.

All in all, this work paves the way for the application of heterostructures based on inverted plasmonic lattices with three-fold symmetry to improve the performance of a wide range of plasmonic sensors and surface enhanced spectroscopies.

## Figures and Tables

**Figure 1 nanomaterials-11-01217-f001:**
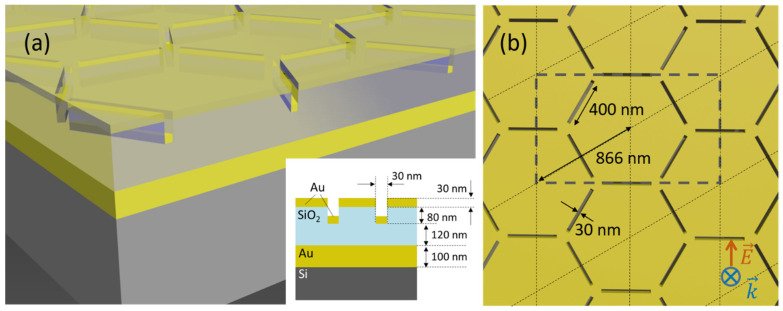
(**a**) Scheme of the heterostructure. The inset shows a schematic cross-sectional view. (**b**) Top view showing the honeycomb lattice, the hexagonal Bravais lattice (dotted lines), a rectangular unit cell (dashed grey lines), and some relevant parameters of the design. Regular rhombus are primitive cells of the hexagonal lattice. The cell for the simulations is the rectangular unit cell. At the bottom left corner of the panel, there is a schematic representation of the electric field and propagation vectors of the incident radiation.

**Figure 2 nanomaterials-11-01217-f002:**
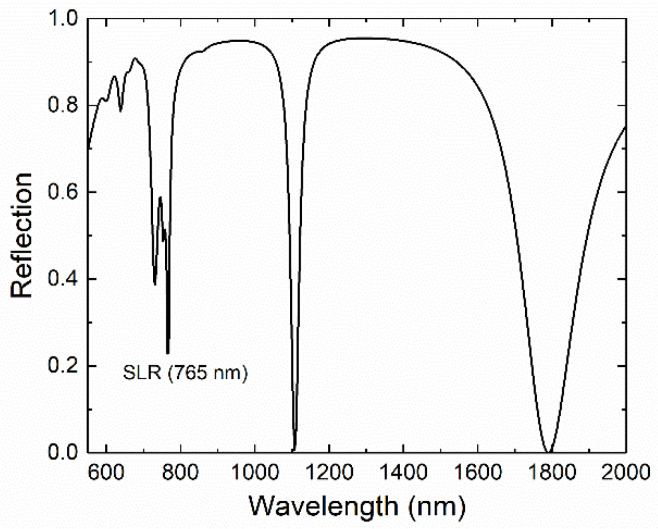
Simulated reflection spectrum as a function of the wavelength of the incident radiation. The sharp peaks at about 765 and 1107 nm correspond to the surface lattice resonance (SLR) of the Au inverted honeycomb lattice at the air–Au and Au–SiO_2_ interfaces, respectively.

**Figure 3 nanomaterials-11-01217-f003:**
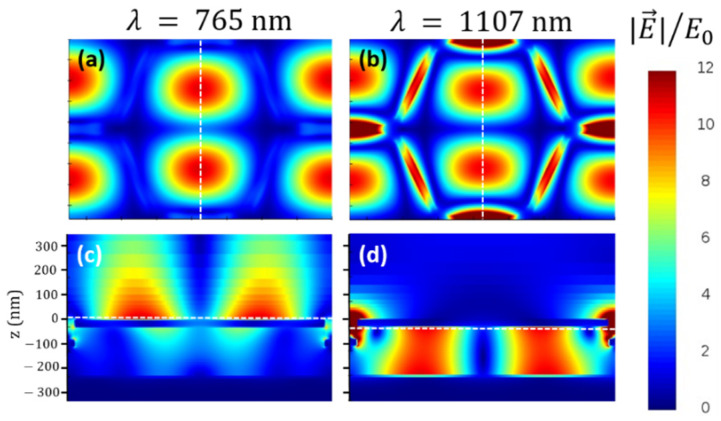
Near-field distributions associated with the SLR at 765 nm (**a**,**c**) and 1107 nm (**b**,**d**). (**a**,**b**) correspond to in-plane views 10 nm above and below the inverted honeycomb lattice, respectively, at the heights indicated by the dashed lines in the corresponding panels (**c**,**d**). (**c**,**d**) depict transversal cross sections of the heterostructure along the dashed lines plotted in the corresponding in-plane view panels (**a**,**b**), respectively. $\left|\vec{E}\right|/E_{0}$ stands for the normalized modulus of the electric field with respect to the modulus of the incident radiation.

**Figure 4 nanomaterials-11-01217-f004:**
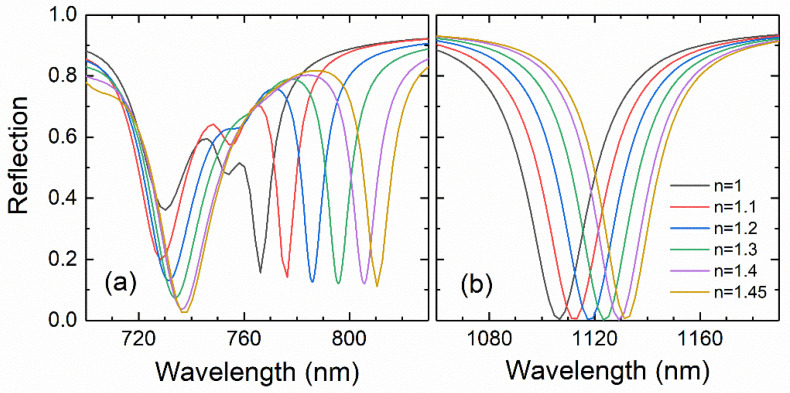
Reflection spectra around the wavelength ranges where the SLRs corresponding to the top (**a**) and bottom (**b**) interfaces of the Au inverted honeycomb lattice take place. The spectra were simulated for several values within 1.1 and 1.45 of the refractive index of a test layer of 50 nm in thickness resting on top of the heterostructure. The spectrum in Figure 2 without the test layer is included as a reference (black line).

**Figure 5 nanomaterials-11-01217-f005:**
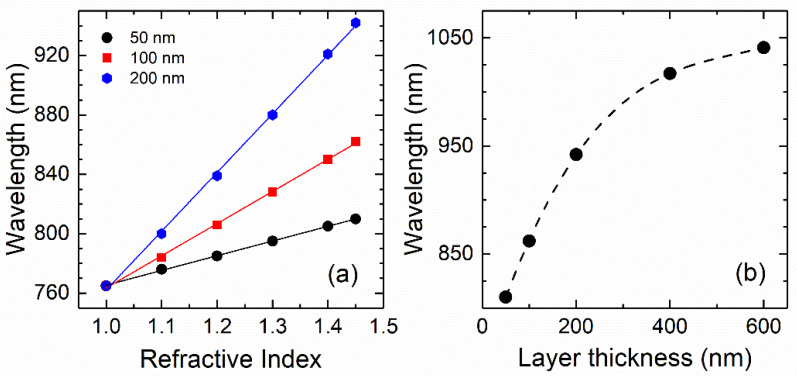
(**a**) Wavelength of the peak corresponding to the SLR at the test layer–Au interface as a function of the refractive index for three values of the test layer thickness, namely, 50, 100, and 200 nm. (**b**) Wavelength of the SLR peak corresponding to the test layer–Au interface for n=1.45 as a function of the layer thickness. The dashed line is only a guide to the eye.

**Figure 6 nanomaterials-11-01217-f006:**
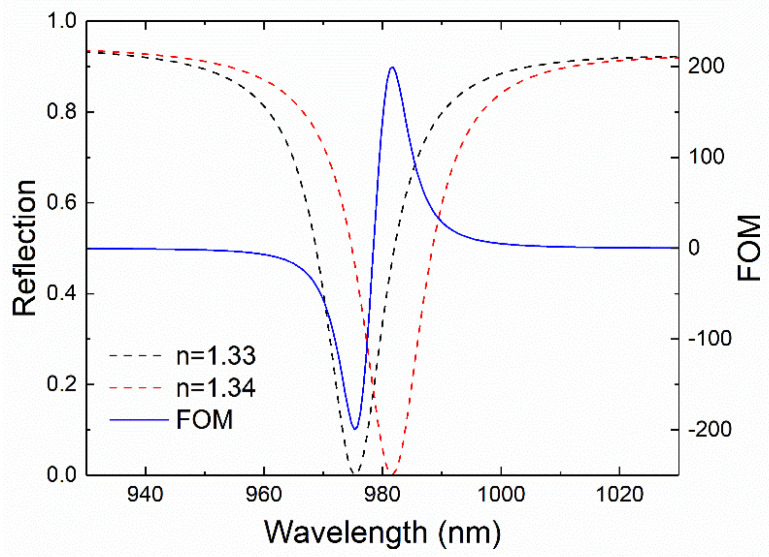
Reflection peak corresponding to the SLR at the test layer–Au interface for a test layer of 600 nm in thickness with values of the refractive index of 1.33 (black dashed line) and 1.34 (red dashed line). The blue solid line corresponds to the computed FOM from these two spectra using Equation (2).

**Figure 7 nanomaterials-11-01217-f007:**
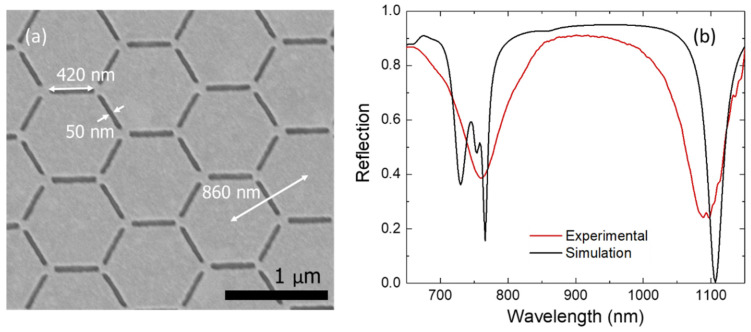
(**a**) SEM image of the manufactured sample. (**b**) Experimental (red curve) and simulated data (black curve) corresponding to the reflection spectrum as a function of the wavelength. The simulated curve is the same as in Figure 2 and is included here for the sake of comparison.

## Data Availability

Not applicable.

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
