# Peer review of "An Inverted Honeycomb Plasmonic Lattice as an Efficient Refractive Index Sensor"

_nanomaterials, 2021, doi:10.3390/nano11051217_

Round 1

Reviewer 1 Report

In this manuscript " An inverted honeycomb plasmonic lattice as an efficient refractive index sensor", the author has evidenced the high performance of the presented heterostructure based on a Au inverted honeycomb lattice for sensing purposes. This demonstration will be quite helpful in sensitivity of plasmonic sensors. Overall, the manuscript is of interest, but there are a few points that have to be revised before the paper can be considered for publication in nanomaterials. 1.In this work, the author presents a sensing application based on the detection of small changes in the refractive index of the medium above the heterostructure by measuring the shift in the wavelength of the SLR at the interface with the top Au layer. Is there a definite value for the measurable minimum change in refractive index of the device proposed in the article? 2.The sharp peaks are different between the experimental and simulation data of the reflection spectrum, In Figure 7, why these differences have little effect on the measurement? 3.The author describes that the simulated curve in Figure 7(b) is consistent with the curve of Figure 2. What is the point of the loss of the peak at 1107nm in the measurement? 4.Some references may help enrich the introduction, e.g., Opto-Electronic Advances 4, 200061 (2021), Advanced Optical Materials, 2020, 8(18): 2000609, and Opto-Electronic Advances 3, 190040 (2020)

Reviewer 2 Report

Rodríguez-Álvarez and co-workers reported the heterostructure based on an Au inverted honeycomb lattice for refractive index analysis. Most simulated results were given in the manuscript. The simulation results were very interesting and meaningful. However, there are always discrepancies between the simulation and the actual situation. It is not convinced that such a structure has the potential to be used for refractive index analysis from only the simulated data. Therefore, I hope the authors can give more real experimental results to verify the simulation results and make the discussion in the manuscript. In the current form, this manuscript is not suitable for publication on Nanomaterials.

  1. The experiments for refractive index analysis should be performed and the analysis procedure should be included in the manuscript. More importantly, please compare and discuss with the simulation results of Figure 5.
  2. The information for the length or width of this structure in the TEM image (Figure 7) should be clearly marked like Figure 1b. By the way, the pitch of the hexagonal lattice in the TEM image is roughly estimated to be 1 uM which is larger than that of the designed structure (Figure 1b). Therefore, if it is because of the difference in structure size that the simulated and experimental results are slightly different. The authors can discuss this point.
  3. In the caption of Figure 7, the authors mentioned “the simulated curve is the same than in Figure 2”. This is not an appropriate statement.

Round 2

Reviewer 2 Report

The authors addressed properly the reviewer's comments. The manuscript is now acceptable for publication.